# Knowledge, Attitudes, and Practices of Fish Farmers Regarding Water Quality and Its Management in the Rwenzori Region of Uganda

Athanasius Ssekyanzi [1,2,*], Nancy Nevejan [1], Ronald Kabbiri [3], Joshua Wesana [2,4,5] and Gilbert Van Stappen [1]

1   Laboratory of Aquaculture & Artemia Reference Center, Department of Animal Sciences and Aquatic Ecology, Ghent University, Coupure Links 653, B-9000 Gent, Belgium
2   Department of Crop & Animal Production, Faculty of Agriculture and Environmental Sciences, Mountains of the Moon University, Fort Portal P.O. Box 367, Uganda
3   Department of Agribusiness & Extension, Faculty of Agriculture & Animal Sciences, Busitema University, Jinja—Malaba Road, Tororo P.O. Box 236, Uganda
4   Food and Markets Department, Natural Resources Institute, University of Greenwich, Central Avenue, Chatham Maritime, Kent ME4 4TB, UK
5   Department of Agriculture Economics, Faculty of Bioscience Engineering, Ghent University, Coupure Links 653, B-9000 Gent, Belgium
*   Correspondence: ssekyanziarthur@gmail.com or athanasius.ssekyanzi@ugent.be; Tel.: +256-773-962-035

**Abstract:** As the number of inhabitants in Sub-Saharan Africa (SSA) increases, demand for animal-source proteins outstrips the current supply. Aquaculture is promoted to sustain livelihood and for improved food security. However, the production in SSA is still low at less than 1% of the total global production. Poor water quality is cited to be one of the factors limiting the growth of the aquaculture sector and is attributed to limited familiarity with standard aquaculture practices. Thus, a knowledge, attitudes, and practices (KAPs) survey was carried out among fish farmers in five districts of the Rwenzori region. Our results showed that 81% and 80% of them had poor knowledge and practices concerning water quality in aquaculture, respectively. Seventy percent did not know that fish farming caused pollution, while 68% believed that there was no need to treat fish farm effluents. Only 45% showed good attitudes towards water quality management. Fish farmers that fed fish with only complete pellets and those that combined them with locally available products (LAP) were 8 and 5 times more likely to possess more knowledge ($p < 0.01$) on water quality as compared to others that used only LAP. Slight improvements in attitudes and practices for every unit increment in knowledge were observed ($p < 0.05$). This limited familiarity with water quality management could severely impede the growth of aquaculture, as well as the sustainable utilization of available water resources. Therefore, there is a need for more training and improvement of extension services among fish farming communities.

**Keywords:** effluents; aquaculture production; rural; land-based; smallholder; sub-Saharan Africa; locally available products (LAP)

## 1. Introduction

The development of the African aquaculture sector has been broadly segmented into 3 phases i.e., the introductory phase (1950–1970), the expansion phase (1970–1995), and the emergence of commercial aquaculture (1995—till today) [1]. The latter phase accounts for a twenty-fold rise in production from 110,200 tons in 1995 to 2,196,000 in 2018, with a compound annual growth rate (CAGR) of 16% [1,2]. This rise is attributed to both the growth of private sector-controlled small and medium-scale enterprises (SMEs), and the development of big commercial enterprises [1].

Ninety-nine percent of the aquaculture production in Africa is from inland freshwater systems, which are dominated by the culture of indigenous species such as Nile tilapia

(*Oreochromis niloticus*) and African catfish (*Clarias gariepinus),* while mariculture only contributes 1% [1–4]. The observed rapid growth of aquaculture production in Sub-Saharan Africa (SSA) is partly attributed to the expansion of cage culture from nine cages in 2006, to more than 20,000 in 2019 [4]. These are mainly located on lakes Victoria (Uganda, Kenya, Tanzania), Kariba (Zambia, Zimbabwe), Kivu (Rwanda, Democratic Republic of Congo; DRC), Muhazi (Rwanda), and Volta (Ghana), which host 91% of the total inland cage culture [4,5].

Despite this fast growth, SSA still accounts for less than 1% of global aquaculture production [4,6]. However, SSA is home to approximately 14% of the total world population [6]. The rapidly growing population compounded by the decline of capture fisheries has led to higher fish demand than the current supply in this region [4,6]. For example, SSA imported 1.5 times more fish products between 2015 and 2019 than was produced from local aquaculture [4]. Factors such as the limited markets, transaction costs, unavailability of quality feed, limited supply of fingerlings, limited availability of suitable land, shortage of fish diseases management expertise, inadequate regulatory frameworks and policies [1,7], poor water quality [8,9], as well as the overall lack of knowledge and skills in fish farm management [10], limit the sector's growth.

The aquaculture growth rate (34% annually) in Uganda is one of the highest in Africa, having grown from 820 tonnes in 2000 to 112,344 by 2017 [6,11]. During this same period, the value of aquaculture production grew from USD 820,000 to 259,121,000, at a rate (40%) higher than the global (9%) and African (7%) averages [6]. This has been attributed largely to the expansion of commercial cage culture on lake Victoria [4,5], and less on rural and/or smallholder fish farming enterprises generally characterized by low and unreliable yields [8,10,12–15]. Many fish farmers using earthen ponds in the country have abandoned the activity as it is deemed unviable [12,13]. Poor water quality has been regularly cited as one of the major problems faced [8,16].

Water quality contributes to the success of any fish farming enterprise [17–20]. Managing water quality involves a proper understanding and manipulation of complex interactions between the stocked organisms and their ecosystem to enhance survival for increased productivity [18,20]. On the other hand, fish farm effluents also negatively impact receiving water systems and the environment [21–24]. This results in the general quality degradation of receiving water systems, eutrophication, increased water treatment costs, and other downstream impacts on the environment [18,20,21,25]. The lack of necessary knowledge and skills in fish farm management is a prevalent challenge among African fish farmers [5,10,14,16,24]. However, the magnitude of these deficiencies among the fish farming communities has never been assessed.

This study aimed at (1) assessing the level of knowledge, attitudes, and practices (KAPs) of fish farmers regarding water quality and its management, and (2) evaluating the relationship between KAPs with socio-demographics, as well as the production factors. Although KAPs studies involving aspects of antimicrobial use and resistance [26], as well as biosecurity [27–29] in aquaculture, have previously been carried out, this study is the first to focus on water quality and its management in rural fish farming communities. The study was carried out in the Rwenzori region because it is one of the areas in Uganda where smallholder fish farming is being practiced but with low recorded yields as reported in previous findings [30].

## 2. Materials and Methods

### 2.1. Study Area

The Rwenzori region is located close to the equator, along the border of Uganda and the DRC [31]. This region is estimated to cover approximately 3.1% of the surface area of Uganda [31,32]. The study was conducted in five districts (Kyegegwa, Kyenjojo, Kabarole, Bundibugyo, and Kasese) of the region. During this study, Bunyangabo was considered to be part of Kabarole district despite having recently been demarcated as a fully standalone administrative district by the Ugandan government (Figure 1).

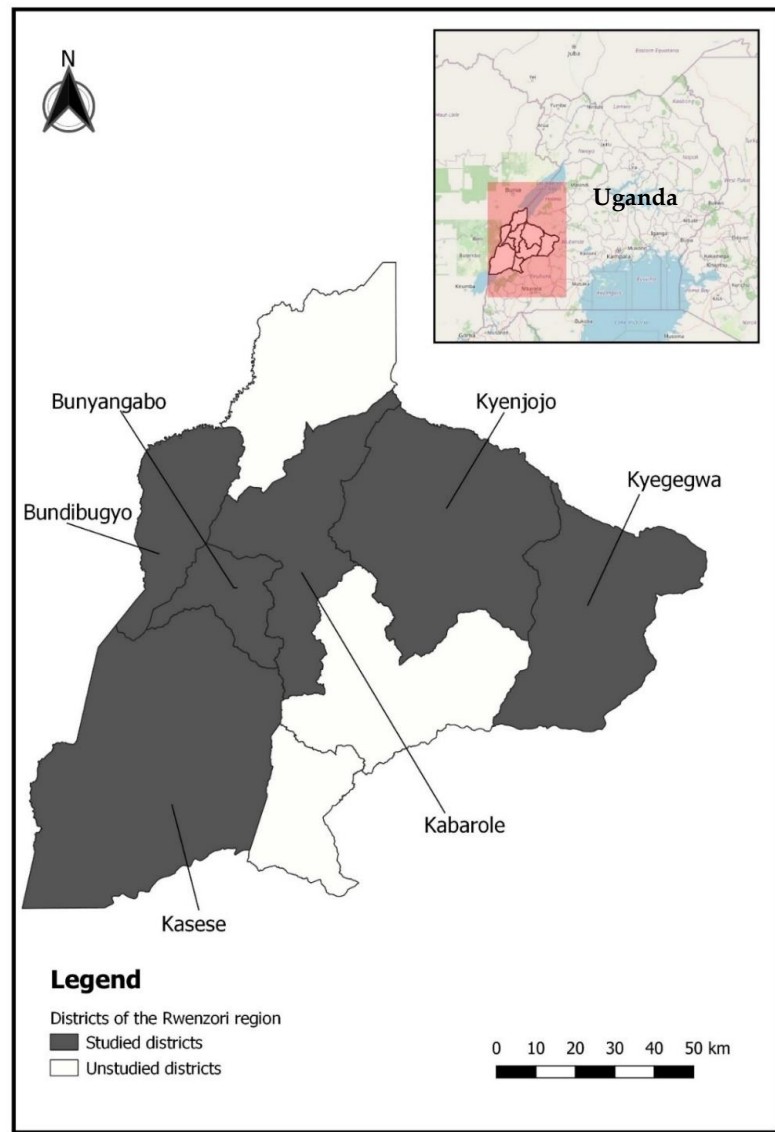

**Figure 1.** Map of the Rwenzori region showing the districts (with their location in Uganda, see map top right) where fish farmers were interviewed (grey) and where no fish farmers were interviewed (white). Bunyangabo is a former county of Kabarole that was established as an independent district in 2017. In this study, those two districts were studied together as Kabarole district where complete fish farmers' lists were still held.

*2.2. Respondent Selection*

A KAP questionnaire was used to source respondents that actively participate in fish farming. Based on the lists of fish farmers obtained from the respective districts' databases, the potential respondent population size for all the districts was 436. At a 95% confidence level and 5% margin error, the minimum sample size of 205 fish farmers was determined using the single proportion estimation method [33]. Under the guidance of local district guides, active fish farmers were identified, having inquired whether they were actively involved in fish farming activities. Thereby, a total of 246 fish farmers (39 from Kabarole, 53 from Kyenjojo, 41 from Kyegegwa, 60 from Kasese, and 53 from Bundibugyo) were randomly taken to participate in the survey interviews. Visits were made to each farm, followed by the administering of interviews.

### 2.3. Data Collection

Responses from the participants were obtained using questionnaires. The research team developed the questionnaire in English. These were translated into the respective local languages (Lutoro, Lwisi, and Lukonjo) for the collection of information from the respondents. A preliminary test of the questionnaire ($n = 13$) was conducted in Kabarole district. The questionnaire was revised and validated with the supervision of senior researchers. The interviews that lasted for approximately 40 min were conducted among selected fish farmers. Section 1.0 of the questionnaire covered information about farm and farmer character-istics. Section 2.0 covered information about fish productivity (species, culture facilities, type of feeds applied, length of the production cycle, yield). Only data from the previous complete production cycle (2019) was included in this section. Section 3.0 was used to collect information about water availability and quality management (water source, water quantity, and source reliability). Section 3.1 was used to collect data about knowledge, attitudes, and practices on water quality management. This section had four subsections. The first subsection (3.1.1) was used to obtain information about the knowledge and awareness of water quality. The second subsection (3.1.2) was used to collect data about the attitudes/perceptions toward water quality monitoring and management. The third subsection (3.1.3) was used to collect information about the practices of water quality monitoring and management. The last (subsection 3.1.4) was used to collect information about incidences of water quality problems. A copy of the questionnaire is provided in the Supplementary Materials. The surveys were conducted between February and March 2021.

### 2.4. Data Management and Analysis

One respondent was omitted due to providing incomplete information. Therefore, 245 respondents were considered for the KAPs analysis. The obtained data were entered into an MS Excel spreadsheet and thoroughly cross-checked for errors. The data were then cleaned and further processed. The procedures followed were similar to what was described by Jia et al. (2017) [29], Pham-duc et al. (2019) [26], and Kambey et al. (2021) [27]. Each validated question was independently analyzed with answers assigned a score, either 1 (correct) or 0 (incorrect). Assessment of whether the responses were correct or wrong was based on the discussions of Boyd & Tucker (1998) [34] and MAAIF (2020) [35] on aquaculture water quality management. To analyze how individual participants performed in each of the knowledge, attitude, and practice categories, the sum of each participant's answers for that section was calculated. Those whose responses were deemed $\geq 75\%$ correct in a given section of the questionnaire were considered to have good knowledge, attitudes, or practices, concerning water quality management. Fish farmers that scored between $\geq 50$ and $< 74\%$ were classified as possessing fair knowledge, attitudes, or practices, while those that scored $< 50\%$ were deemed to have poor levels of the same [27,29]. The associations between attitude/practice levels with socio-demographics, production factors, as well as yield, were assessed using ordinal logistic regression. The dependent variables (attitude and practice level) were measured at three ordinal levels i.e., good, fair, and poor. The independent variables were district (Bundibugyo, Kabarole, Kasese, Kyegegwa, and Kyenjojo), gender (male, female), age (34 and below, 35 and above), experience (two years or less, above two years), level of education (primary or lower, secondary or higher), culture method (monoculture, polyculture), sources of information (one or none, two or more), information is obtained from other persons (yes, no), knowledge score, and attitude score. The validity of the model was assessed using the model fitting information ($p < 0.05$), goodness-of-fit tests (Pearson and Deviance) with $p > 0.05$, and test of parallel lines ($p > 0.05$). Due to 344 (66.7%) of the cells of the dependent variable "knowledge levels by subpopulations" having zero frequencies, this data could not fulfill the assumptions required to run an ordinal logistic regression. Therefore, counts of respondents with fair and good knowledge of water quality were combined to form a single category known as "Fair to good knowledge. Then, assessing the factors related to the knowledge levels of fish farmers concerning water quality and its management (Section 3.5.1) was carried

out using binary logistic regression due to having only two categories of the dependent variable ("Poor" and "Fair to good knowledge levels"). The validity of the latter regression model was also assessed for the goodness of fit using both Pearson and Deviance chi-square statistics ($p > 0.05$), while Pseudo R-Square (Nagelkerke) was used to assess the strength of the association. The Omnibus Test of Model coefficients ($p < 0.05$) and Hosmer and Lemeshow Test ($p > 0.05$) were also used. All the statistics were carried out using IBM SPSS statistics 28.0.1 software.

## 3. Results

### 3.1. Participant Demographics

Most of the interviewed fish farmers were male (84%) as compared to 16% who were female (Table 1). The respondents were evenly distributed between age groups ranging from 15–24, up to >55 years old. Only 2.9% of the interviewed fish farmers had no form of formal education. The results show that the majority of the fish farming enterprises were individually owned (84%) compared to 6% being part of a company, and 7% being owned by community-based organizations. Of the interviewed fish farmers, 25% had less than two years of fish farming experience, while the rest had farmed fish longer. Among the 224 fish farmers that were part of individually owned enterprises, 74 (33%) were classified as low-income (poor), while 150 (67%) were middle-income or higher (Table 1).

### 3.2. Fish Production Characteristics

Most of the fish farms (88%) were into grow-out aquaculture, while others complimented it with fish fry production (10%). Monoculture (78%) was the most practiced culture method as compared to polyculture (22%), with Nile tilapia (70%) being the most farmed fish species, followed by African catfish (27%), and then mirror carp *Cyprinus carpio* (2%). Some fish farmers (1%) reported farming other species such as *Tilapia zillii*, and other unidentified types of catfishes that had either invaded their culture facilities from the wild or that they had intentionally introduced. Earthen ponds were the most utilized culture facilities (91%) (Table 2). Less than 25% of the fish farmers fed their fish on commercial pellets (either floating or sinking), but the majority used locally available products (LAP) such as vegetables (30%) and homemade feed (22%). Examples of common homemade feeds included maize bran, cooked plantain (banana), and sweet potato peelings. Fourteen percent of the fish farmers fed fish on leftover food scraps, while 5% only fertilized water to grow plankton as natural food. On the other hand, 1% of the fish farms did not administer any form of feeding, while 4% applied other forms of feed such as industrial waste products from breweries (Table 2).

**Table 1.** Socio-demographic characteristics of interviewed active fish farmers in the five districts of the Rwenzori region of Uganda.

| Item | No. of Responses (%) |
|---|---|
| *Gender* | |
| Male | 205 (83.7) |
| Female | 40 (16.3) |
| *Age (years old)* | |
| 15–24 | 22 (9.0) |
| 25–34 | 41 (16.7) |
| 35–44 | 76 (31.0) |
| 45–54 | 62 (25.3) |
| >55 | 44 (18.0) |

**Table 1.** *Cont.*

| Item | No. of Responses (%) |
|---|---|
| *Highest education* | |
| No formal education | 9 (3.7) |
| Primary education | 84 (34.3) |
| Secondary education | 81 (33.1) |
| Tertiary education | 70 (28.6) |
| Other (short course or training from CBOs/NGOs) | 1 (0.4) |
| *Category of farm* | |
| Individual fish farm | 210 (85.7) |
| Private sector/Company | 14 (5.7) |
| Government agency/parastatal | 1 (0.4) |
| University/training institute/college | 2 (0.8) |
| Civil society/association/group | 18 (7.3) |
| *Years of fish farming experience* | |
| <2 | 62 (25.3) |
| 2–3 | 88 (35.9) |
| 4–5 | 40 (16.3) |
| >5 | 55 (22.4) |
| *Income of farmers (n = 224)* | |
| Low income/poverty | 74 (33.0) |
| Middle-income and above | 150 (67.0) |
| *Contribution of fish farming to income (n = 224)* | |
| <25% | 131 (58.5) |
| 25–50% | 56 (25.0) |
| 51–75% | 35 (15.6) |
| >75% | 2 (0.9) |

Note: CBO: Community-based organization; NGO: Non-governmental organization. These usually conduct training to empower smallholder fish farmers in rural areas. The frequencies of the income level of farmers and the contribution of fish farming to income were obtained from 224 respondents. The 23 missing respondents belonged to government agencies/parastatals, universities/institutes/colleges, or civil society/associations or groups.

### 3.3. Water Resources Characteristics

Streams/rivers (29%), swamps (35%), and groundwater/wells (26%) were the main sources of water for fish farming. Municipal piped water and rainfall were also utilized as water sources by 1% and 4% of the fish farmers, respectively. Eighty percent of the respondents reported carrying out some form of water exchange at their fish farms. While 87% claimed that the water quantity was reliable for fish farming throughout the year, 9% disagreed, and 4% were not sure. The majority (66%) noticed changes in water quality during the production season. Incidences of algal blooms (63%), change in turbidity (22%), effects of decomposition such as off-odors (7%), and variation of physicochemical parameters from the optimal (8%), were the main water quality problems outlined. It was claimed by 34% of the fish farmers that most water quality issues occurred in the dry seasons, while 28% reported experiencing such incidences during the rainy periods. Others (31%) reported such incidences occurring unexpectedly at any time, while a few (7%) were not sure. Most of the respondents (67%) mentioned water exchange as the remedy to these water quality anomalies. While 13% of the respondents that noticed water quality deterioration claimed to do nothing when such issues arose, others transferred

the fish to other facilities (2%), practiced feed management (2%), applied manual removal of dirt and wastes (6%), as well as consulted with the aquaculture/fisheries officers (1%). A few (1%) reported that they abandoned production when water quality issues arose. However, 72% of these farmers reported having never experienced any fish losses due to problems associated with water quality. Most of the respondents (57%) operated constant flow-through systems. Approximately 22% had well-designed culture systems with proper water inlets and outlets, while 20% did not practice any water exchange (Table 3).

**Table 2.** Fish production characteristics of selected fish farms in districts of the Rwenzori region.

| Item | No. of Responses (%) |
|---|---|
| *Fish production activities* | |
| Grow out | 244 (87.8) |
| Hatchery | 29 (10.4) |
| Recreation | 5 (1.8) |
| *Culture method* | |
| Monoculture | 191 (78.0) |
| Polyculture | 54 (22.0) |
| *\* Cultured fish species* | |
| Nile tilapia (*Oreochromis niloticus*) | 218 (69.9) |
| African catfish (*Clarias gariepinus*) | 84 (26.9) |
| Mirror carp (*Cyprinus carpio*) | 6 (1.9) |
| Other (*Tilapia zillii*, and unidentified wild catfishes) | 4 (1.3) |
| *\* Culture facilities* | |
| Earthen ponds | 224 (91.1) |
| Lined ponds | 7 (2.8) |
| Concrete tanks | 14 (5.7) |
| Cages | 1 (0.4) |
| *\* Feeds applied* | |
| Commercial extruded floating pellets | 98 (20.3) |
| Commercial sinking pellets | 18 (3.7) |
| Homemade feed (maize bran, etc.) | 105 (21.7) |
| Vegetables | 147 (30.4) |
| Food-scraps | 68 (14.1) |
| Green water/fertilization | 24 (5.0) |
| No feeding | 5 (1.0) |
| Other (local brewery waste etc.) | 18 (3.7) |

Note: * Implies that the frequencies are obtained from multiple responses.

### 3.4. Knowledge, Attitudes, and Practices on Water Quality Management in Fish Farms

3.4.1. Knowledge of Water Quality Management

Ten percent of the respondents had never heard about water quality issues. Friends and family members (31.3%) were the commonest sources of information about water quality for the respondents who reported having ever heard about the subject of "water quality". This was followed by fisheries or extension officers (26.5%). Other sources of information were also used as shown in Table 4.

**Table 3.** Water resources and water quality dynamics of the fish farms in the Rwenzori region.

| Item | No. of Responses (%) |
|---|---|
| *\* Water sources* | |
| Rainfall | 18 (6.5) |
| Groundwater/well | 72 (25.9) |
| Stream/river | 82 (29.1) |
| Lake | 7(2.5) |
| Swamp | 98 (35.3) |
| Municipal/piped water | 2 (0.7) |
| *Water source reliability* | |
| Yes | 212 (86.5) |
| Not sure | 10 (4.1) |
| No | 23 (9.1) |
| *Water quality changes observed* | |
| Yes | 162 (66.1) |
| No | 83 (33.9) |
| *Types of water quality changes observed (n = 162)* | |
| Turbidity | 36 (22.2) |
| Algal blooms | 102 (63.0) |
| Physicochemical parameters | 13 (8.0) |
| Off-odors | 11 (6.8) |
| *The season when water quality problems are observed (n = 162)* | |
| Dry season | 55 (34.0) |
| Rainy season | 46 (28.4) |
| Anytime | 50 (30.9) |
| Not sure | 11 (6.8) |
| *Remedies to poor water quality (n = 162)* | |
| Administering fertilizers | 15 (9.3) |
| Water exchange | 109 (67.3) |
| Transfer fish to another culture facility | 3 (1.9) |
| Feed management | 3 (1.9) |
| Manual removal of dirt and wastes | 9 (5.6) |
| Abandon production | 1 (0.6) |
| Consult with the fisheries officer | 1 (0.6) |
| No action | 21 (13.0) |
| *Fish losses due to water quality problems (n = 162)* | |
| Yes | 47 (27.8) |
| No | 117 (72.2) |
| *Water exchange routine* | |
| No water exchange | 50 (20.4) |
| Constant flow-through | 140 (57.1) |
| Controlled inlet and outlet | 55 (22.4) |

Note: \* Implies that the frequencies are obtained from multiple responses.

**Table 4.** Frequencies of sources of information on water quality management among fish farmers of the Rwenzori region, Uganda. These frequencies were obtained for multiple responses. CBO stands for community-based organizations.

| Source of Information on Water Quality | No. of Responses (%) |
|---|---|
| TV | 11 (3.0) |
| Radio | 28 (7.7) |
| Internet | 9 (2.5) |
| Academic journal | 6 (1.7) |
| School/college/university | 8 (2.2) |
| Government agencies | 39 (10.7) |
| Friends/family members | 115 (31.7) |
| CBO training | 13 (3.6) |
| Fisheries/extension officer | 94 (26.4) |
| Never heard of it | 38 (10.5) |

Above 80% of the fish farmers that were interviewed did not know what good or poor water quality for fish farming was (Figure 2). Only seven percent identified that good water quality for fish farming should possess favorable levels of parameters such as pH, dissolved oxygen (DO), temperature, ammonia, and turbidity for fish. The others considered good quality water for fish farming to be green in color, visibly clean, or transparent. Six percent of the respondents either did not know and/or either linked quality to the smell of water or the absence of physical particles. Only fourteen percent of respondents knew that bad water quality for fish farming meant water with unfavorable abiotic parameters for cultured fish. The majority of fish farmers responded that poor quality water is one "that is unclear" (18%), "visibly dirty" (30%), "badly colored" (17%), or "muddy" (7%). Others differed from all these responses by saying that poor water quality meant stagnant water or water with physical foreign bodies such as plastic and rubbish (Figure 2).

Most of the respondents (75%) did not know the importance of regularly testing water quality in their fish farming facilities. They either assumed that the quality of water should never be tested, it should be tested only when fish are sick, exhibit poor response to feeding, or whenever the technical person (extension officer) visits. Specifically, 29% of the respondents were unaware of the necessity to test water quality. Only 62 (26%) of the respondents knew that water quality testing should be a regular activity at the fish farm (Figure 2).

Although only 4 (2%) of the fish farmers perceived all fish farming systems to be prone to water quality deterioration, a large majority (98%) believed their fish culture facilities were less likely to experience poor water quality at any stage of the production cycle. Furthermore, 70% of the respondents also did not know that fish farming activities led to the addition of pollutants into the natural water systems and environment. Furthermore, 173 (68%) did not see the need of treating fish farm effluents before releasing them back into the environment (Figure 2).

However, 82% of the respondents knew that the quality of water influenced the yields. Furthermore, 60% of the respondents acknowledged that water quality influenced the fish species one could rear, preferring African catfish that can withstand turbid water over Nile tilapia. Approximately 65% of the respondents also knew that the quality of feed could influence water quality. It was also established that 97% of the respondents knew at least one technique for maintaining good water quality at a fish farm, while 80% were aware of a method to improve water quality once deterioration occurs. However, water exchange was the most reported remedy in both scenarios. Most respondents (70%) acknowledged the importance of keeping water quality records at a fish farm (Figure 2).

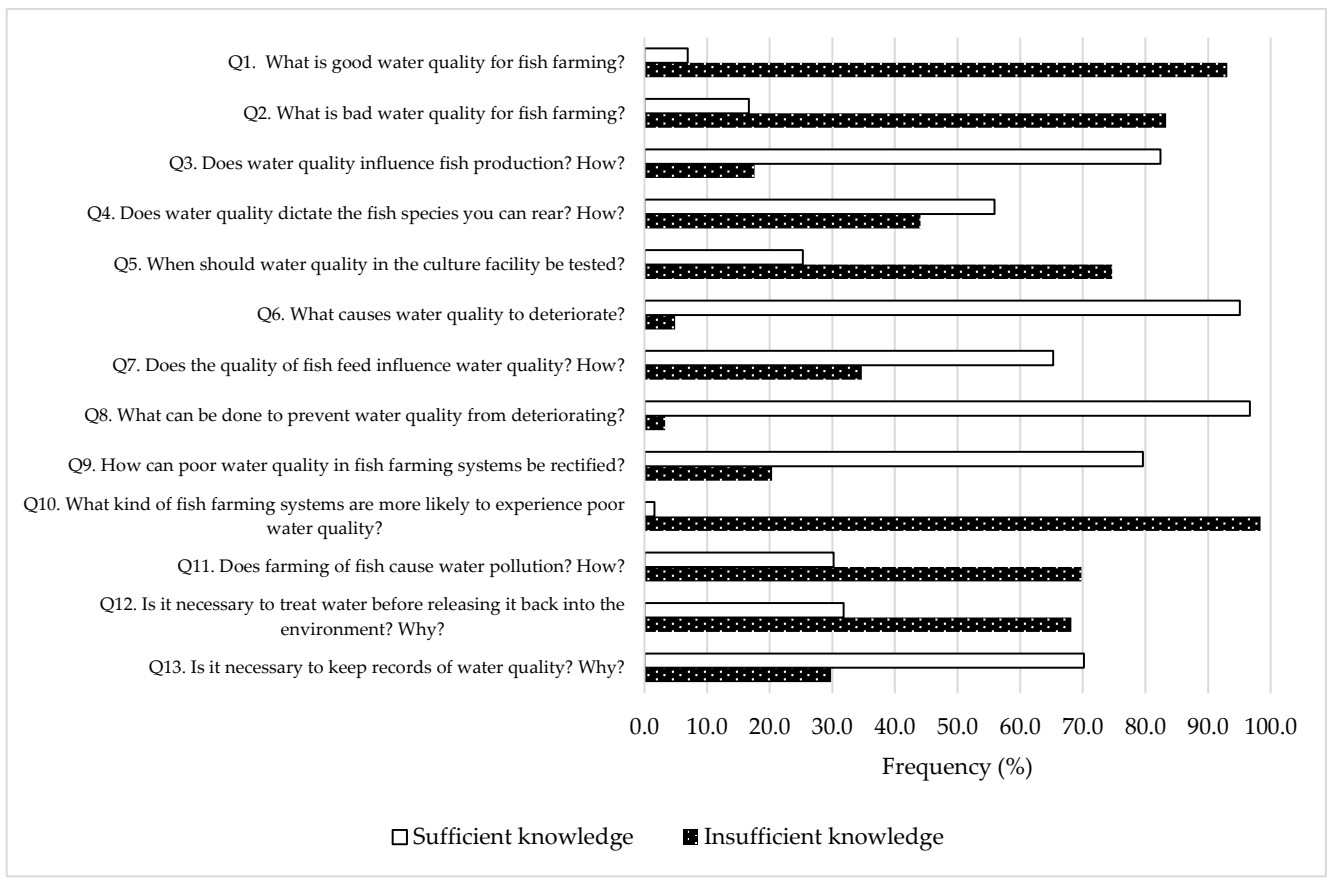

**Figure 2.** Percentage frequencies of correct (sufficient knowledge) and incorrect (insufficient knowledge) responses to questions regarding knowledge about water quality and its management among fish farmers of the Rwenzori region, Uganda.

3.4.2. Attitudes of Fish Farmers toward Water Quality Monitoring and Management

Most of the fish farmers exhibited positive attitudes toward water quality and its management (Figure 3). However, respondents were unwilling to invest in water quality management perceiving it to be costly (70%), considered treating effluents as unimportant (56%), and did not perceive any negative effect of effluent on the environment (59%) (Figure 3).

3.4.3. Water Quality Management and Monitoring Practices among Fish Farmers

The water quality management and monitoring practices among the fish farmers were generally poor. More than three-quarters (76%) of the respondents never tested the water quality of the source before setting up their fish farm. Most respondents (69%) never considered the quality of water when selecting fish species to culture. While more than half of the respondents (54%) claimed to be monitoring water quality at their fish farms, 81% of them neither monitored a single water quality parameter nor knew what to monitor. Most of the fish farmers (79%) had no access to water quality testing kits. However, 16.3% of them reported observing fish behaviour using their eyes or feeling senses (hands) to determine good or poor water quality. Of the 245 respondents, 212 (87%) never tested any water quality parameter during the production cycle, while 90% did not keep any water quality records. Most fish farmers (77%) did not carry out any form of fish farm effluent treatment before releasing it back into the environment. Those who did treat the effluent used it for irrigation (7%), had treatment ponds (1%), or directed it back to a wetland (2%) (Figure 4).

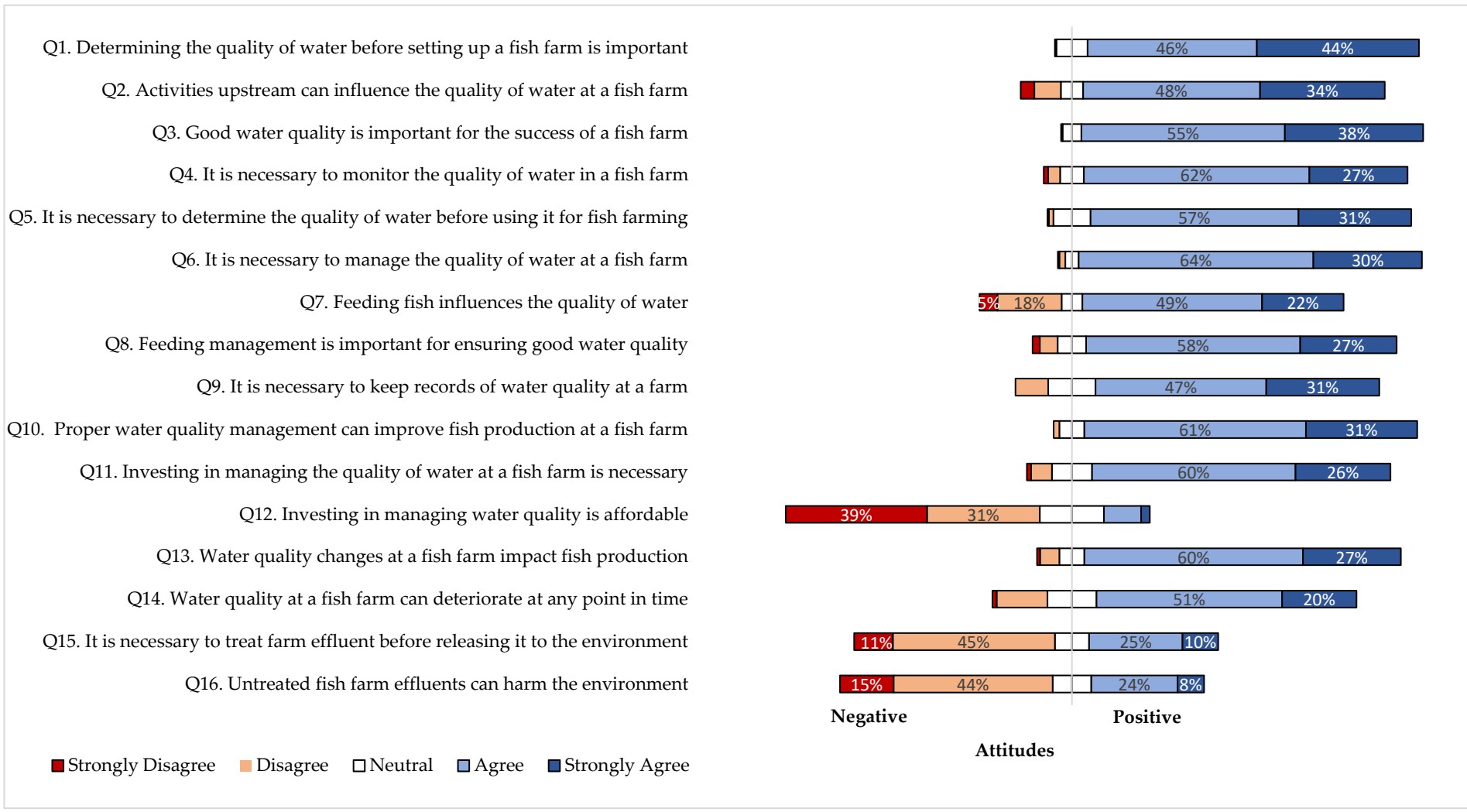

**Figure 3.** Assessment of the attitudes of fish farmers towards water quality and its management in five districts of the Rwenzori region.

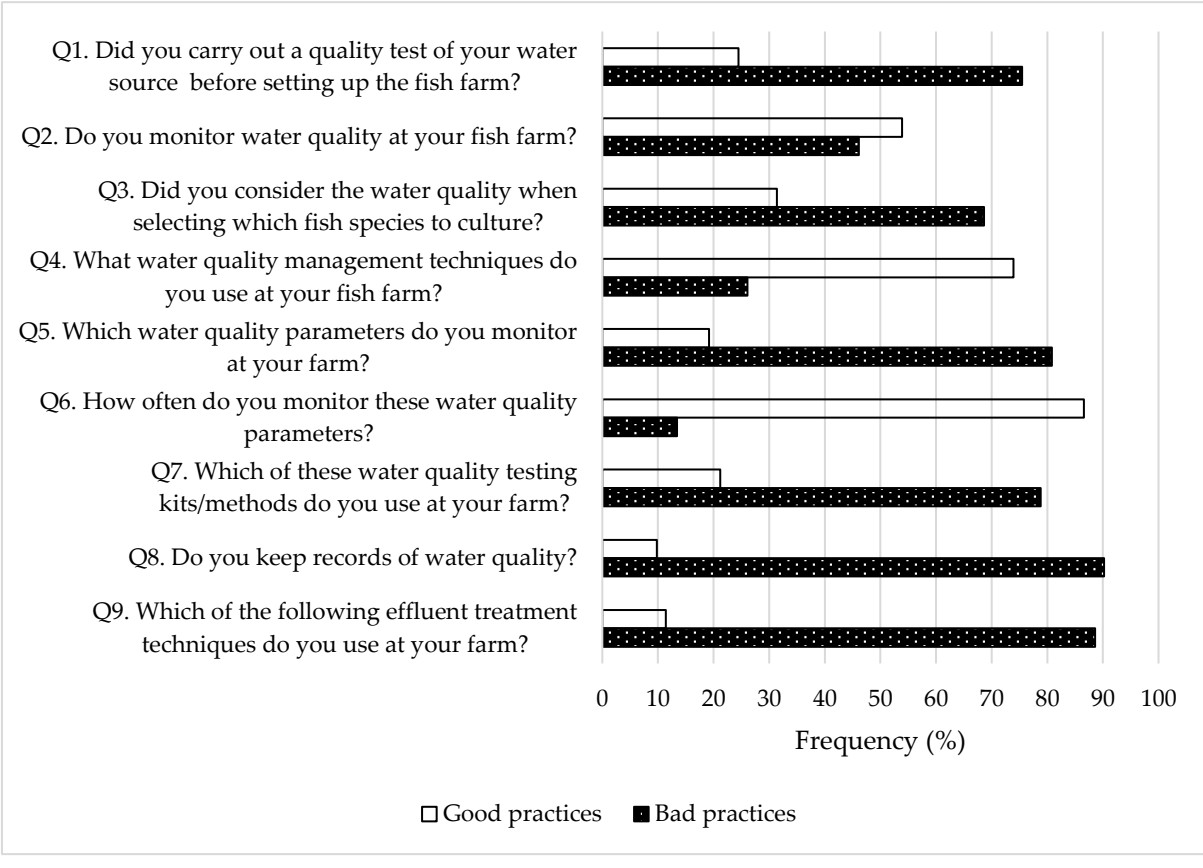

**Figure 4.** Percentage frequencies of correct (good practices) and incorrect (bad practices) responses to questions about practices regarding water quality and its management among fish farmers of the Rwenzori region.

With regards to water quality management techniques employed in aquaculture production, water exchange (29%) was most used, followed by fertilization (28%). Fifty-three (17%) of the 245 respondents claimed not to employ any form of water quality management. Other water quality management techniques such as biofiltration (1%), mechanical filtration (7%), liming (4%), and aeration (9%) were also utilized (Table 5).

**Table 5.** Water quality methods employed by fish farmers in the studied districts of the Rwenzori region.

| Water Quality Management Method | No. of Responses (%) |
|---|---|
| None | 53 (16.7) |
| Biofilters | 3 (0.9) |
| Mechanical filters | 22 (6.9) |
| Fertilization | 89 (28.0) |
| Water exchange | 93 (29.2) |
| Liming | 14 (4.4) |
| Heating | 2 (0.6) |
| Aeration | 29 (9.1) |
| Other (adding ash, picking rubbish, etc.) | 12 (4.1) |

Note: These results were obtained from multiple responses.

*3.5. Factors Related to Knowledge, Attitudes, and Practices Regarding Water Quality Management in Fish Farms*

3.5.1. Factors Related to Fish Farmers' Knowledge of Water Quality Management

According to the computed knowledge level scores, 81%, 17%, and 2% of the fish farmers had poor, fair, and good knowledge about water quality and its management, respectively. Binary logistic regression was performed to establish the effects of socio-

demographic factors (gender, age, education), production factors (experience, fish culture methods, applied feed types), information sources, and yields on the level of knowledge about water quality. Most of the assessed factors showed no significant association with knowledge. However, fish farmers that applied only complete pellets and those that combined them with LAP were 8 and 5 times more likely to possess more knowledge ($p < 0.01$) on water quality as compared to those that used only LAP. Though there was a positive relationship between fish yield and the level of knowledge on water quality management, this was not significant (Table 6).

**Table 6.** Binary logistic regression analysis of factors associated with knowledge with regards to water quality management among fish farmers of the Rwenzori region.

| | OR (95% CI) | *p* Value |
|---|---|---|
| *Farm ownership* | | |
| Private/CBO/Government owned | . | . |
| Individually owned | 1.838 (0.436–7.757) | 0.407 |
| *Gender* | | |
| Male | . | . |
| Female | 0.353 (0.081–1.538) | 0.166 |
| *Age range* | | |
| 34 and below | . | . |
| 35 and above | 0.815 (0.287–2.315) | 0.702 |
| *Completed education level* | | |
| Primary school or lower | . | . |
| Secondary school or higher | 0.941 (0.359–2.469) | 0.902 |
| *Experience in fish farming* | | |
| 2 years or less | . | . |
| 3 years or more | 1.185 (0.477–2.943) | 0.715 |
| *Culture methods* | | |
| Monoculture | . | . |
| Polyculture | 0.632 (0.208–1.916) | 0.418 |
| *Feeds applied* | | **0.003** |
| Locally available products (LAP) | . | . |
| Commercial pellets | **8.319 (2.036–33.998)** | **0.003** |
| Commercial pellets + LAP | **4.940 (1.733–14.082)** | **0.003** |
| *Sources of information on water quality* | | 0.118 |
| None | . | . |
| One or less | 0.311 (0.024–4.037) | 0.372 |
| Two or more | 0.113 (0.007–1.713) | 0.116 |
| *Information from other persons* | | |
| No | . | . |
| Yes | 2.812 (0.230–34.393) | 0.418 |
| Yield per m$^2$ | 1.140 (0.987–1.316) | 0.074 |

Note: Values in boldface indicate are significant ($p < 0.1$). The dependent variable (knowledge level) was transformed into a binary where "fair to good knowledge" was coded as 1 and "poor knowledge" was coded as 0. Fair to good knowledge is the reference category.

### 3.5.2. Factors Related to Fish Farmers' Attitudes towards Water Quality Management

According to the computed attitude scores, 15%, 40%, and 45% of the respondents had poor, fair, and good attitudes toward water quality management, respectively. Ordinal logistic regression was performed to establish the effects of socio-demographic factors (gender, age, education), production factors (experience, fish culture methods, applied feed types), information sources, knowledge score, and yields on the attitudes towards water quality among fish farmers. Male fish farmers were 2.6 times more likely to possess good attitudes toward water quality and its management as compared to the females ($p < 0.05$). However, yield (kg) per square meter dropped by 27% for every unit increase in attitude towards water quality and its management ($p < 0.01$). On the other hand, a positive association was observed between the percentage knowledge score and the attitude levels. That is, for every unit increase in percentage knowledge level score, the attitude levels slightly improved by 1.1 times ($p < 0.01$) (Table 7).

**Table 7.** Ordinal logistic regression analysis of factors associated with attitudes towards water quality management among fish farmers of the Rwenzori region.

|  |  | OR (95% CI) | *p* Value |
|---|---|---|---|
| Threshold | Poor attitudes | 3.886 (0.768–19.664) | 0.101 |
|  | Fair attitudes | 62.468 (11.120–350.942) | 0.000 |
| *District where the farm is located* |  |  |  |
| Bundibugyo |  | 1.078 (0.396–2.939) | 0.883 |
| Kabarole |  | 1.058 (0.333–3.359) | 0.924 |
| Kasese |  | 2.758 (0.926–8.220) | 0.069 |
| Kyegegwa |  | 2.133 (0.724–6.287) | 0.170 |
| Kyenjojo |  | 1 | . |
| *Farm ownership* |  |  |  |
| Private/CBO/Government-owned |  | 1.534 (0.587–4.011) | 0.383 |
| Individually owned |  | 1 | . |
| *Gender* |  |  |  |
| Male |  | **2.596 (1.063–6.342)** | **0.036** |
| Female |  | 1 | . |
| *Age range* |  |  |  |
| 34 and below |  | 0.601 (0.264–1.368) | 0.225 |
| Above 34 |  | 1 | . |
| *Completed education level* |  |  |  |
| Primary school or lower |  | 0.589 (0.296–1.173) | 0.132 |
| Secondary school or higher |  | 1 | . |
| *Experience in fish farming* |  |  |  |
| 2 years or less |  | 1.395 (0.702–2.773) | 0.342 |
| Above 2 years |  | 1 | . |
| *Culture methods* |  |  |  |
| Monoculture |  | 0.943 (0.415–2.141) | 0.888 |
| Polyculture |  | 1 | . |
| *Sources of information on water quality* |  |  |  |

**Table 7.** *Cont.*

|  | OR (95% CI) | *p* Value |
|---|---|---|
| One or none | 0.689 (0.324–1.467) | 0.334 |
| More than one | 1 | . |
| *Information from other persons* |  |  |
| No | 1.020 (0.401–2.597) | 0.966 |
| Yes | 1 | . |
| Yield per m$^2$ | **0.732 (0.600–0.894)** | **0.002** |
| % Knowledge score | **1.097 (1.066–1.129)** | **0.000** |

Note: Values in boldface indicate are significant (*p* < 0.05).

3.5.3. Factors Related to Fish Farmers' Practice of Water Quality Management

The computed practice level scores showed that 80%, 13%, and 7% had poor, fair, and good water quality management practices, respectively. Ordinal logistic regression was performed to establish the effects of socio-demographic factors (gender, age, education), production factors (experience, fish culture methods, applied feed types), information sources, knowledge score, attitude score, and yields on the practice levels of water quality management among fish farmers. Respondents from Bundibugyo were found to be six times more likely to practice good water quality management as compared to the reference (Kyenjojo) when all other variables in the model are kept constant (*p* < 0.05). On the other hand, respondents from company/government/civil society-owned fish farms were 3 times more likely to practice good water quality management as compared to those from individually owned ones (*p* < 0.05). It was also observed that the level of good practice concerning water quality management minimally improved by 1.04 times for every unit increment in percentage knowledge level about the same subject (*p* < 0.05) (Table 8).

**Table 8.** Ordinal logistic regression analysis of factors associated with practices of water quality management among fish farmers of the Rwenzori region.

|  |  | OR (95% CI) | *p* Value |
|---|---|---|---|
| Threshold | Poor practices | 484.758 (20.784–11306.098) | 0.000 |
|  | Fair practices | 2105.520 (81.993–54067.960) | 0.000 |
| *District where the farm is located* |  |  |  |
| Bundibugyo |  | **5.609 (1.311–23.996)** | **0.020** |
| Kabarole |  | 2.431 (0.440–13.423) | 0.308 |
| Kasese |  | 2.981 (0.741–11.988) | 0.124 |
| Kyegegwa |  | 1.226 (0.266–5.644) | 0.793 |
| Kyenjojo |  | 1 | . |
| *Farm ownership* |  |  |  |
| Private/CBO/Government-owned |  | **3.274 (1.112–9.637)** | **0.031** |
| Individually owned |  | 1 | . |
| *Gender* |  |  |  |
| Male |  | 1.008 (0.301–3.374) | 0.990 |
| Female |  | 1 | . |
| *Age range* |  |  |  |

**Table 8.** *Cont.*

|  | OR (95% CI) | *p* Value |
|---|---|---|
| 34 and below | 0.759 (0.256–2.243) | 0.617 |
| Above 34 | 1 | . |
| *Completed education level* |  |  |
| Primary school or lower | 1.565 (0.652–3.760) | 0.316 |
| Secondary school or higher | 1 | . |
| *Experience in fish farming* |  |  |
| 2 years or less | 0.812 (0.343–1.922) | 0.635 |
| Above 2 years | 1 | . |
| *Culture methods* |  |  |
| Monoculture | 0.423 (0.169–1.059) | 0.066 |
| Polyculture | 1 | . |
| *Sources of information on water quality* |  |  |
| One or none | 1.711 (0.649–4.513) | 0.278 |
| More than one | 1 | . |
| *Information from other persons* |  |  |
| No | 0.936 (0.273–3.210) | 0.917 |
| Yes | 1 | . |
| *Yield per $m^2$* | 1.049 (0.879–1.250) | 0.598 |
| % Knowledge score | **1.043 (1.007–1.080)** | **0.019** |
| % Attitude | 1.033 (0.991–1.076) | 0.122 |

## 4. Discussion

Water quality is the most important aspect of any aquaculture operation [17,18,36,37]. It involves skillful manipulation and management of the water ecosystem to ensure survival and increased yields [18,37]. Therefore, it is imperative to have an understanding of the dynamic complex interactions that occur between the stocked organisms and their ecosystem [18]. As pure water quality is rarely found in nature [17], it is imperative to ensure proper monitoring and management in aquaculture facilities.

This study showed that the majority of fish farmers in the Rwenzori region lack sufficient knowledge and hardly practice any water quality management. Less than a quarter of fish farmers showed fair to good knowledge and practice levels while 10% had never heard about water quality and its management in fish farming. More than three-quarters of the respondents neither knew which water quality parameters to test nor had any access to water quality testing kits. A few of them claimed to solely rely on their senses such as sight, smell, and touch to check water quality. However, the quality of water in fish farming facilities is not constant but rather dynamic. Physico-chemical parameters such as dissolved oxygen (DO), pH, carbon dioxide, total ammonia nitrogen (TAN), temperature, turbidity, hardness, and salinity, constantly change depending on various culture conditions and the environment [17,37]. These are influenced by weather, stocking density, time of the day (sunlight), primary production (phytoplankton density and species composition), quality of the feed, cultured fish species, the concentration of organic matter, upstream activities, as well as the source of culture water itself [37,38]. Effective knowledge and skills of how to ensure that these do not deviate from optimum are necessary for the success of any fish farming venture.

Earthen ponds are the most utilized fish culture facilities in the Rwenzori region. In these systems, DO significantly varies during the 24-h duration, being at the lowest level

just before dawn and highest in the late afternoon [17,37]. Maintaining DO above 3.0 mg/L is recommended for warm-water fish species (*O. niloticus* and *C. gariepinus*) to prevent chronic stress and poor response to feeding, high feed conversion ratios (FCR), as well as high susceptibility to diseases [37]. The high variations in environmental temperatures and altitude in the Rwenzori region [31,32] influence the abiotic water quality parameters such as DO and water temperature [38,39], hence justifying the need for routine monitoring and management for higher yields. Also, every fish species has a different upper and lower limit, as well as an optimum range for the respective water quality parameters [35,40]. These ranges also vary with the developmental stage of the fish [40]. However, 69% of the fish farmers confirmed to have not considered water quality when selecting which fish species to culture.

More than three-quarters of the respondents never carried out quality tests on the water source before setting up their fish farms. Some of the commonly reported water sources for aquaculture were swamps (35%) and groundwater/wells (29%), which are usually characterized by low DO levels [35]. Groundwater is also known to usually contain a high concentration of $CO_2$ which is known to negatively affect cultured fish, and at times lead to mortalities [37,41]. Several fish farmers that constructed their earthen ponds in wetlands had no control over water quality since these were filled by seepage. They reported facing problems such as low water levels during dry seasons as well as water quality problems such as the overgrowth of algal blooms and the development of off-odors. On the other hand, the water quality of streams is influenced by the activities upstream such as domestic use, crop and animal husbandry, as well as other economic activities [42,43]. In the Rwenzori region, common artisanal economic activities such as alcohol brewing and palm oil processing are important causes of water pollution in streams (Figures S1 and S2). In districts like Bundibugyo, the streams usually flow through dense forests of cocoa plantations. The decomposing leaves in the water source impact its physicochemical parameters such as pH, DO, dissolved organic matter (DOM), and BOD [44–46] which could negatively impact the fish culture conditions.

More than half of the respondents operated constant flow-through systems (Table 3). Although these systems are efficient at maintaining good water quality [47], they have a downfall of leading to variations in temperatures within the culture facilities. This could be the case in the Rwenzori region where many streams are fed by melting glaciers from the mountains and also have densely forested banks [32,48]. Thus, their water is relatively cold. Therefore, it is good practice to hold water still in an earthen pond especially during the daytime hours to ensure maximum temperatures and primary production due to solar radiation [49]. Water temperatures influence all biological and chemical processes in an aquaculture operation [35,37]. Since fish are poikilotherms, the temperature of water directly influences metabolic rates, thus their appetite as well as response to feeding, stress levels, and growth rates [35,37]. Temperature also influences the minimum DO levels that are safe for fish survival as well as the solubility of oxygen in water, which increases as temperature decreases and vice versa [17,37,50]. Unlike what we observed, it is good practice to monitor daily temperatures in fish farming facilities. For example, it is not advisable to feed fish during cold periods when fish exhibit low appetites and poor responses to feeding [35,50]. The lack of this knowledge as well as the recommended necessary practice towards such a parameter contributes to the wastage of uneaten feed and high FCR [35]. In addition to the deterioration of water quality, this will also lead to economic losses if the farmer uses commercial pellets as feed, which are already known to constitute over 60% of the production costs if used as the sole feed [51].

Although most fish farmers never monitored water quality, a small minority of them reported using organic fertilizers such as cow dung, goat, and chicken droppings as a way to enhance primary production (Table 2) or as a remedy to water quality deterioration (Table 3). On top of that, locally available low-quality products such as vegetables and household leftovers (Table 2) were some of the most administered feed types. The decomposition of the applied low-quality feeds and organic fertilizers consumes oxygen from the water

column [17,52]. Although fertilization is a recommended practice for improved primary production in earthen ponds [53–55], it is important to consider the amount of oxygen consumed over 12 and 24 h ($BOD_{0.5}$ and $BOD_1$) by each specific organic fertilizer to be applied [56]. This is important in determining the perfect timing for applying fertilizers to avoid oxygen consumption by decomposing microbes [17], especially at night when oxygen is at its lowest [37].

Furthermore, the application of poor quality food products (Table 2) such as home-made feed (maize bran, cooked plantain, and potato peels), vegetables (yam leaves, jack fruit, and avocadoes), as well as leftover food scraps leads to low appetite and poor response to feeding [57]. This is because fish possess a quality dietary insight and thus do not consume anything that is provided to them [58]. On top of that, low-quality feeds tend to have a low digestibility [59] which in turn increases the excretion/defecation rates of cultured fish. Such conditions contribute to the loading of the culture water column with nitrogenous and other wastes that result from the decomposition of uneaten feed, as well as excretion [37,38,60,61]. Nitrogenous wastes lead to high stress levels, low growth rates, high FCR, and in the worst scenario, fish mortalities, as well as increased susceptibility to diseases [17,37,51]. In aquaculture, the build-up of nitrogenous wastes in fish farming systems is unavoidable. However, skillful manipulation of the culture environment such as the appropriate proliferation of bacteria (pH between 7–9; temperature approximately 24–29 °C), water exchange, feeding, and stocking density management are known to keep the concentration of toxic ammonia in check [37,60,62].

Our findings further show that 7% of the 162 fish farmers report occurrences of off-odors in their fish ponds. Off-odors can only be corrected by draining the pond and exposing the bottom to fresh air [37]. This is a common phenomenon in both earthen and lined ponds, where the decomposition of organic matter at the bottom leads to the production of hydrogen sulfide ($H_2S$), especially under anoxic conditions [37]. $H_2S$ gas is characterized by the presence of rotten egg odor and is extremely toxic to fish when seined or disturbed [37]. Therefore, it is important to avoid any detectable odors [37] by practicing good water quality management.

Changes in water turbidity were also one of the commonly experienced water quality anomalies (22%) observed by fish farmers. Turbidity is another water quality aspect that influences the welfare of fish in aquaculture facilities whose tolerance levels vary among species [18,37]. High levels of turbidity are usually caused by phytoplankton, suspended, and dissolved solids [37]. The occurrence of algal blooms (63%) was also the most reported water quality anomaly (Table 3). Though a high density of phytoplankton signifies good levels of primary productivity in a culture system, very high densities could lead to anoxia at night due to the respiration and decomposition of dead algae [17]. Some of these suspended and dissolved materials are also known to cause off-flavors in fish [37].

More than a third (34%) of the respondents reported having never noticed any water quality changes. This could be due to the majority of them lacking sufficient knowledge about water quality. These observed low knowledge and practice levels exhibited could be the reason for the reported low yields that characterize Ugandan smallholder fish farming enterprises [8,13]. It could also be one of the major reasons contributing to the reported phenomenon where many smallholder pond fish farmers in Uganda are abandoning the activity due to deeming it unviable [13].

The results from this KAPs study also showed that the majority of the respondents lacked the knowledge that fish farming activities led to the addition of pollutants into the natural water systems and environment (Figure 2). They also saw no need to treat fish farm effluents before releasing them back into the environment (Figure 3). On top of that, over three-quarters of the fish farmers (77%) reported not to be treating fish farm effluents in any way before releasing them back into the environment (Figure 4). However, culture water is enriched with nitrogen, phosphorus, organic matter, and suspended solids during feeding, excretion, and fertilization [18,20,21,25]. For example, it has been proved that only 20 to 40% of nitrogen and phosphorus that is applied to ponds through feeding is

recovered during harvest with much of the rest being lost into the system or effluent [25]. Though they may be added in low amounts, these pollutants gradually accumulate, thus negatively impacting the receiving environments [18,20,21,25,51]. In rural sub-Saharan Africa, freshwater is a highly coveted resource that is usually utilized communally for all production and domestic purposes [63,64]. It is a hinge for survival and food security [64]. Therefore, it is necessary to implement some form of effluent treatment in rural smallholder fish farms to ensure communal harmony and sustainability of the freshwater resources.

The observation of males being more likely to possess good attitudes towards water quality and its management as compared to females (Table 7) can be explained by the fact that rural and smallholder fish farming enterprises in Rwenzori and other areas of SSA are characterized by being male dominated [30,65,66]. Therefore, the males likely possess more knowledge of aquaculture in general which they obtain from their friends/family members, identified in our results as the main source of information on water quality. Yields were also observed to drop as attitudes toward water quality management improved. This could be a result of fish farmers that experience lower yields being more aware of the importance of water quality in fish rearing, thus having more positive attitudes towards the subject. On the other hand, respondents from private company/CBO/government-owned fish farming settings were three times more likely to practice good water quality management as compared to those from individually owned ones. This could be due to the availability of resources for the former group to access skilled labour, train their staff/members, and purchase water quality testing equipment as compared to the latter group.

Unlike Asia which has had a long tradition of technical experiences in aquaculture that dates back several millennia [67], fish farming is less than a century old in most parts of SSA. On top of that, it is still facing challenges of poor extension, inadequately trained middle-level labor force, poor logistics, and most especially little scientific application [68] as proved in our study. Also, the subsistence nature of most of the fish farming operations plus the prevalent absence of routine management [68] could be one of the major causes of the observed poor yields. Therefore, training fish farmers in the field of water quality and its management is necessary just as it was observed that knowledge was positively associated with attitudes and practices.

## 5. Conclusions

The findings of this study show low levels of knowledge and practices of water quality management among smallholder fish farmers in the Rwenzori region. These deficiencies in knowledge and practice are a major impediment to achieving the targets of increasing aquaculture production, poverty alleviation, and food security through the promotion of fish farming in the region. On top of that, the fish farmers' poor attitudes towards effluent management are a threat to sustainable water resource use. Yet, fresh water is a highly coveted natural resource that is important for survival and socio-economic development.

**Supplementary Materials:** The following supporting information can be downloaded at: https://www.mdpi.com/article/10.3390/w15010042/s1, Figure S1: Artisanal palm-oil processing facilities are next to streams that are water sources for fish farms. These drain their waste directly into the streams; Figure S2: An artisanal alcohol brewing facility (**A**) and (**B**) clothes are being washed in the flowing stream. The questionnaire used during this study is also provided.

**Author Contributions:** Conceptualization, A.S., R.K., J.W., N.N. and G.V.S.; methodology, A.S., R.K. and J.W.; validation, A.S.; formal analysis, A.S. and J.W.; investigation, A.S.; data curation, A.S. and R.K.; writing—original draft preparation, A.S.; writing—review and editing, A.S., R.K., J.W., N.N. and G.V.S.; visualization, A.S.; supervision, N.N. and G.V.S.; project administration, G.V.S.; funding acquisition, N.N. and G.V.S. All authors have read and agreed to the published version of the manuscript.

**Funding:** This research was funded by VLIR-UOS (funding agency) for DGD (Belgian Government), grant number UG2019IUC027A103.

**Institutional Review Board Statement:** Not applicable.

**Informed Consent Statement:** Verbal informed consent to participate in the study was sought from each participant before being interviewed.

**Data Availability Statement:** The data that support the findings of this study are available from the corresponding author, [SA], upon reasonable request.

**Acknowledgments:** We acknowledge the fisheries officers of the respective districts for their active participation and guidance during this study. The authors also thank the research assistants for their service during the study.

**Conflicts of Interest:** The authors declare no conflict of interest.

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
