# Peer review of "Knowledge, Attitudes, and Practices of Fish Farmers Regarding Water Quality and Its Management in the Rwenzori Region of Uganda"

_water, doi:10.3390/w15010042_

Round 1
Reviewer 1 Report
I found this manuscript well-written and accurate, it could represent a good base for the topic in the treated area. Some minor suggestions to improve its quality.
Line 55: double-check the references style (same as 110, 157, 465, 476, 511-512)
Sections 4 and 5: please fix as 4. Discussion and 5. Conclusion
Line 406: especially when placed in natural confined environments, see for example and use for better argue https://doi.org/10.3390/w14010108
References: double-check the style and italicize scientific names
Best regards
The Reviewer
Author Response
Dear Reviewer,
We are grateful for your review and eye-opening comments. We did our best to respond to all of them. Attached is our response letter.

Reviewer 2 Report
The Introduction should be rearranged and summarized in order for the text to logically lead from the statistical data, through the identified problem to the goal of the research. For example, move section line 58 - 69 behind the first section (behind line 46); section 79 - 89 behind line 57 and after it place section 90 -101.
Material and methods are correctly presented, and the research is very well conceived and carried out. The statistical analysis is very comprehensive and almost all relative statistical tests were used. I'm not sure that such an extensive statistical analysis is necessary for a useful discussion and conclusions (but I would keep it anyway).
Results are well presented, including tables and graphs. Some data are described and presented in tables and graphs, so the text of the results could be a little shortened in this sense.
The discussion must focus on the objectives and results of the research. Most of the text includes theorizing about the basic parameters of water quality, and this should be significantly shortened, and discussion about the results and goals of the research should be expanded.
The conclusions are satisfactorily presented and come from the objectives and results of the research.
Author Response
Dear Reviewer,
We are grateful for your review and eye-opening comments. We totally agree to all your comments and did our best to address all of them. Attached is our response letter. Thank you!
Athanasius Ssekyanzi (on behalf of all authors)

Reviewer 3 Report
This MS assessing the level of knowledge, attitudes, and practices of fish farmers regarding water quality and its management, considering socio-demographics and production factors in Sub-Saharan Africa, Rwenzori region of Uganda. Having in mind the importance of water quality management for successful production – this MS presents a good example that could performed in other regions for knowledge gaps detection in order to increase production, improve product quality and profitability of the sector.
Title - appropriate
Abstract – well written and adequate
Introduction - Missing fluency
Authors are talking about production, then production systems, then water quality, and after that again about production…. Paragraphs are not connected adequately, and this is a reason why this chapter should be improved.
Materials and methods – well described, in detail.
Results – well written
Discussion – should be improved. There is to many basic teaorethical fact that should be removed (a lot of text looks like a review paper on basic water quality parameters). This chapter should be shorten, but fluently concentrated on the main goal of the research!
Author Response
Dear Reviewer,
We are grateful to your review and eye-opening comments. We agree with all your comments and did our best to address all of them. Thank you for your contribution to our work.
Athanasius Ssekyanzi (on behalf of all authors)

Round 2
Reviewer 3 Report
MS is imprived, and can be accepted.